# Cyclic Homomorphic Encryption Aggregation (CHEA)—A Novel Approach to Data Aggregation in the Smart Grid

Daniel Sousa-Dias [1], Daniel Amyot [1,*], Ashkan Rahimi-Kian [1,2], Masoud Bashari [2] and John Mylopoulos [1]

1. School of Electrical Engineering and Computer Science, University of Ottawa, Ottawa, ON K1N 6N5, Canada; dsous064@uottawa.ca (D.S.-D.); ashkan.kian@iemssolution.com (A.R.-K.); jm@cs.toronto.edu (J.M.)
2. IEMS Solution Ltd., Communitech, Kitchener, ON N2G 1H6, Canada; masoud.bashari@iemssolution.com
* Correspondence: damyot@uottawa.ca; Tel.: +1-613-562-5800 (ext. 6947)

**Abstract:** The transactive energy market is an emerging development in energy economics built on advanced metering infrastructure. Data generated in this context is often required for market operations, while also being privacy sensitive. This dual concern has necessitated the development of various methods of obfuscation in order to maintain privacy while still facilitating operations. While data aggregation is a common approach in this context, many of the existing aggregation methods rely on additional network components or lack flexibility. In this paper, we introduce Cyclic Homomorphic Encryption Aggregation (CHEA), a secure aggregation protocol that eliminates the need for additional network components or complicated key distribution schemes, while providing additional capabilities compared to similar protocols. We validate our scheme with formal security analysis as well as a software simulation of a transactive energy network running the scheme. Results indicate that CHEA performs well in comparison to similar works, with minimal communication overheads. Additionally, CHEA retains all standard security properties held by other aggregation schemes, while improving flexibility and reducing infrastructural requirements. Our scheme operates on similar assumptions as other works, but current smart metering hardware lags in terms of processing power, making the scheme infeasible on the current generation of hardware. However, these capabilities should quickly advance to an accommodating state. With this in mind, and given the results, we believe CHEA is a strong candidate for aggregating transactive energy data.

**Keywords:** aggregation; homomorphic encryption; smart grid

## 1. Introduction

Advanced metering infrastructures (AMI) have enabled new functionalities in the power delivery sector. Smart meters enable frequent collection of detailed energy profile data from the homes of consumers [1]. Additionally, smart energy grids facilitate the transmission, storage, and analytic computations of consumers' energy profile data by means of Information and Communication Technology (ICT) networks that inter-operate with energy grids. Novel developments of intelligent demand response by means of smart thermostats, smart plugs, smart lighting, and smart appliances, as well as optimal power flow algorithms, AI-based electric and thermal/cooling loads predictions are examples of technological progresses in the smart energy grid sector [2]. These progressions will enable greater energy economic efficiency through models such as transactive energy (TE) markets, further enabling high frequency peer-to-peer energy trading between prosumers (consumers who also produce energy) and power utilities [3]. These models will also contribute to greater energy efficiency, due to shorter electrical transmission distances as well as improved load balancing, demand response, and AI-assisted predictions.

Although a standardized transactive energy market (TEM) platform has yet to arise, innovative research works and proposals for TEM platforms are being published regularly, as recently surveyed by Garcia et al. [4]. These platforms often employ blockchain technology

to secure private information such as consumer trading history, identifying information, and account balances, among others [5]. However, an often-overlooked privacy concern is that of energy usage data. Usage data is regularly required for critical operations such as automated demand response, power flow optimization, and billing. While it is not inherently identifying, many researchers believe it should fall under the category of protected private information [6]. This is due to the fact that this data can facilitate many malicious attacks on consumers, including profiling, trading pattern recognition, and malicious trading [7].

As energy usage data is required for many critical functions, it cannot be obfuscated entirely, and instead must be accessible in some form to the distribution systems operator (DSO). Proposals for adapting this information have included adding noise to the data, performing algebraic transformations that preserve large scale statistical properties, and using battery banks to disguise a household's true energy needs [8–10]. However, the approach we believe to be the most effective is aggregation [11]. Data aggregation is successfully used in many other industries, including in healthcare [12]. It has the benefit of not changing the data (like noising and algebraic transformation would), as well as not requiring additional equipment to be installed at the household (like battery-based techniques would).

Many protocols have been proposed to facilitate data aggregation in smart grids. However, as will be seen in the related work, these protocols suffer from drawbacks in terms of hardware requirements and/or flexibility. Additionally, these protocols were not designed with transactive energy in mind.

Our proposed protocol, named *Cyclic Homomorphic Encryption Aggregation (CHEA)*, was designed from the ground up to be integrated into a transactive energy market environment. It is designed to be both flexible and distributed, which are features that make it a natural complement to blockchain-based TEM models.

In this paper, we outline the design of the CHEA protocol, investigate its strengths and weaknesses from a security standpoint, examine the simulation results of the protocol, and compare its performance with other energy data aggregation schemes.

The key contributions of CHEA, introduced in this paper, include:

1. Network infrastructure: While blockchain-enabled TE networks and smart grids often have auxiliary nodes such as fog nodes (FNs) or edge nodes, CHEA does not rely on them. This enables networks that do not have auxiliary nodes to use our scheme without installing additional hardware. Networks that already have such nodes will still benefit, as these nodes will have more resources available to them to perform other tasks (e.g., blockchain validation) than if they were involved with aggregation.

2. Key distribution: Many proposed aggregation techniques in this space rely on complicated cryptographic key distribution schemes in order to function. This is because data in this context must be aggregated regularly, requiring synchronization between the data generating, requesting, and aggregating parties. In contrast, other cases—such as medical data—might only need to be aggregated once for a study. Our scheme sidesteps the issue by performing the aggregation process entirely locally to the group of data owners.

3. Single points of failure: Trusted authorities (TA) and FNs can be single points of failure from both a security and operational standpoint. If an FN goes offline, then its region of smart meters will no longer be able to report to the DSO. If a TA goes offline then the entire network will not be able to report data. If either are compromised then an adversary would be able to acquire individual readings from consumers, enabling the attacks discussed in Section 2. Our scheme does not rely on either of these mechanisms.

4. Scalability: It is noted by Ming et al. [13] and others that the resources of TAs could become overwhelmed if the area covered by a DSO became too large. Since key generation happens locally in CHEA, this concern does not exist. Additionally, as a neighbourhood grows, new aggregating nodes would need to be installed to support additional meters (in schemes that require them). Our scheme is not subject to this

scaling factor. The distributed nature of the protocol means that scaling is limited only by the communication capacity of the DSO.

5. Decentralization: As discussed in [7] the trend in TEM development is towards decentralized applications. A decentralized solution for data aggregation supports this trend and confers the same benefits seen by turning other power management mechanisms into distributed applications.

6. Flexibility: CHEA supports flexible group membership, group size (which corresponds to aggregated data resolution), and locality, meaning that groups can be sparse or dense. These variables allow the scheme to support diverse applications that might have different data requirements.

7. Fault tolerance: On-the-fly group generation means that member smart meters are guaranteed to be active, provided that they do not malfunction during the ∼500 ms aggregation cycle.

This paper will be of interest to practitioners who are designing Transactive Energy Markets (TEMs), smart meters, and smart grid (SG) infrastructure. It provides a novel solution to a common problem in these domains, making it directly useful to SG and TEM operators. Smart meter hardware designers should consider these solutions to inform the technical specifications of the meters. In addition, researchers can use our results as a basis for the comparison of novel schemes, or they may choose to build on our scheme or adapt it for other application fields.

The rest of this paper is structured as follows. Section 2 presents the background and motivation behind the design and implementation of an innovative cybersecurity protocol for smart grids and transactive energy markets' stakeholders. Section 3 outlines similar research works in the field and discusses the various approaches to smart grid data aggregation that have been considered. In Section 4, the CHEA protocol is described in detail. In Section 5, a formal security analysis is performed to demonstrate the privacy-related benefits of the proposed approach. In Section 6, the implemented CHEA protocol is demonstrated in a simulated smart grid network environment. We also discuss the simulation results, as well as how they compare to simulations of similar schemes. Section 7 discusses limitations of our experiments. Finally, in Section 8, we conclude the study with a reflection on the results and suggestions for further research.

## 2. Background and Motivation

Transactive energy markets (TEMs) are meant to represent autonomous distributed platforms for trading energy and reserves among prosumers of energy (with numerous and different types of distributed energy resources—DERs) and distribution system operators (DSOs). Built on top of smart grid infrastructure, a TEM consists of software that facilitates the trading of energy directly between consumers, prosumers, and DSOs. This capability has the potential to increase efficiency, lower costs, and reduce environmental impacts [14,15].

A typical TEM architecture consists of power consumers equipped with smart meters (SMs), prosumers with distributed energy resources, battery energy storage systems (BESS), electric vehicles (EVs), and a traditional power generation station [15–17]. Additionally, there will be some form of distributed system operator, sometimes referred to as control center (CC) or cloud control center (CCC), who handles billing, data management, demand response, demand prediction, and other critical maintenance and operational tasks [16,18].

There is no shortage of data generated within smart grids and transactive energy markets. While some of the mentioned data are personal, such as a user's transaction history, identity, or credit standing, other sources are more opaque.

CHEA is concerned with the privacy protection of energy usage data. These data are required for many critical functions:

- Physical operation of the power grid, for example, when performing state estimation;
- Novel functionality offered by the smart grid, such as automated demand response;
- Operating a transactive energy market, for billing, transaction verification, etc.

A recent literature review that examined privacy concerns in various transactive energy market implementations found that, while steps were generally taken in TEM proposals to protect identifying information, energy usage data are often mishandled [7].

Several studies, including those from McDaniel and McLaughlin [6] and from Lisovich et al. [19], note that energy usage data should be considered private data, and that their leakage can lead to a number of undesirable outcomes. These can include revealing appliance profiles in consumer households, spying, facilitating theft, and allowing hackers to understand activity within the home [5,6,19].

While proposed TE models have neglected this area of concern, smart grid research has produced a number of novel methods for making such data available for important functions, while maintaining user privacy [5]. Additionally, existing methods of aggregating data have been adapted for smart grid environments.

Some proposed methods include:

- *Algebraic Transformation* [8]: Algebraic transformation refers to a family of mathematical techniques that enable the modification of a set of data so that the individual values are altered, but the results of certain computations remain consistent.
- *Battery Filtering*: Battery filtering is a method of disguising usage data proposed by Kalogridis et al. [9]. This method suggests using a battery energy storage system in the home (this could be an EV, a Tesla Powerwall, or another BESS product) as a buffer between the home and the power grid. The BESS would be discharged to meet the short term-term energy demands of the home and charged regularly from the grid. The use of such an energy buffer has the effect of obfuscating the real-time energy usage patterns in the home while maintaining power availability.
- *Data Aggregation* [11]: Commonly used in the medical sciences [12], data aggregation is the process of summarizing data for analysis. Generally, this will consist of performing a summation of each dimension or feature of the data before transmitting them to the data receiver. This has the benefit of allowing analysis of real data without compromising the privacy of individuals; since only the aggregate is analyzed, the data receiver cannot tie any individual measurement to a specific person.
- *Random Noise* [10]: The introduction of random noise is a common method of providing access to data while preserving privacy and preventing statistical attacks. This involves generating noise, or random datapoints, using a random function, such as Perlin noise [20]. Noise can have different statistical properties, such as smooth transitions between points. The generated noise is then used to modify the real measured values (for example, by adding the absolute value of the noise at a particular point) enough to disguise them from attackers, but subtly enough that the result is still useful to the party analyzing the data. In circumstances where approximations suffice, this can be an appropriate solution.

Table 1 displays the attributes of the privacy-preserving data collection methods discussed. **MA1** is hardware independence; **MA2** is preservation of exact measurements; **MA3** is discrete values reporting; and **MA4** is timely data reporting (for use in demand response applications).

**Table 1.** Comparison of privacy-preserving data collection methods.

| Method | MA1 (HW) | MA2 (Exact) | MA3 (Discr.) | MA4 (Timely) |
|---|---|---|---|---|
| Algebraic Transformation | yes | no | yes | yes |
| Battery Filtering | no | no | yes | no |
| Data Aggregation | yes | yes | no | yes |
| Random Noise | yes | no | yes | yes |

Our scheme (CHEA) performs data aggregation in order to preserve privacy. We find this to be most appropriate for the TE environment for several reasons. Unlike introducing random noise and algebraic transformation, aggregation allows the DSO to work with real

values. While some operations—such as demand response—might function adequately given approximate values, others—for example, transaction verification—are not so forgiving. Battery filtering is a compelling solution, but suffers from requiring prosumers to own expensive hardware. User privacy should not be predicated on purchasing additional equipment, and for this reason, a software solution is preferable. Of course, users with EVs or BESS can still employ battery filtering if they desire.

Homomorphic encryption (HE) is a mechanism commonly employed to facilitate privacy-preserving data aggregation [21]. HE refers to encryption in which mathematical operations can be performed on the cyphertext, and the result will be equivalent to having encrypted the result of the same operation performed on the plaintext. For example, assuming an HE-based encryption function $E()$ and the use of the sum operator $(+)$ as the aggregation function, then $E(x) + E(y) = E(x + y)$. Hence, the aggregated result can be decrypted without knowing the nature of the individual operands.

While there are many aggregation schemes based on HE, simulation results (presented later in Section 5) indicate that CHEA excels uniquely in a few key areas (on top of being a "truer" distributed application), including privacy, collusion-resistance, and flexibility, which make it a valuable contribution to the field.

## 3. Related Work

There is extensive body of literature on the topic of data privacy. Even if we restrict ourselves only to a smart grid context, many solutions exist, as hinted at in Section 2. In this section, we focus our analysis of similar works on privacy solutions that employ data aggregation, and ideally those that are also based on homomorphic encryption. A search on Scopus (the most comprehensive index for such literature) with the query privacy AND ("smart grid" OR "transactive energy") AND "data aggregation" AND "homomorphic encryption", executed on 1 December 2023, returned 100 journal and conference papers. The related work closest to ours is discussed below.

There are many privacy-preserving aggregation schemes that rely on homomorphic encryption [13,21–26]. Of those, the majority [22–25] use the Paillier cryptosystem [27,28], which we also chose to use when implementing CHEA due to its relative computational efficiency. Other HE cryptosystems, such as Elliptic Curve ElGamal [29,30], are occasionally used and some researchers compared the performance of multiple cryptosystems [31].

The vast majority of aggregation protocols [13,26,32–37] use a system in which smart meters connect to an intermediate node who aggregates the data before sending them to the distributed system operator. This architecture is outlined in Figure 1.

There are several drawbacks to this architecture. One is that each aggregating node represents a single point of failure, whose outage would prevent a large region of smart meters (the ones aggregated by the failing node) from reporting data. Another is that collusion between the DSO (who owns the HE private key) and an aggregating node is trivial and hence may compromise many households. Finally, there are concerns of scalability given the need to add physical nodes to support neighbourhood expansion.

Another common theme is the use of a trusted authority (TA) to generate and distribute cryptographic keys. This can leave the system vulnerable to a number of attacks, including man-in-the-middle, false data injection, and data deletion if the TA is compromised [13,21,22]. Key distribution is a central problem for traditional schemes, such that it has become its own area of research. For example, Cheng et al. [38] propose a key distribution scheme that reduces communication overhead and improves security. One can thus conclude that avoiding key distribution altogether would be a valuable proposition.

Some solutions do attempt to address these concerns. For example, the scheme by Chen et al. [25] attempts to improve resilience and flexibility by enabling variable subsets of meters, but still relies on a TA to generate and distribute keys.

Instead of relying on external fog nodes, several of the aggregation schemes use in-network aggregation, or aggregation that happens on the smart meters as the data are sent along. This means that intermediate meters will add their encrypted measurements

to those sent by previous meters, a strategy we also use in CHEA. In fact, one of the earliest examples of a smart-grid aggregation scheme for data privacy used this technique. Li et al. [11] describe a technique in which an aggregation "tree" is created to describe the path the data will take. In their case they are employing in-network aggregation for the sake of computation and communication efficiency, so the aggregation tree remains static based on the network topology (as it would not make sense to send the data down a less efficient path). Despite making use of this technique for a different purpose than CHEA, it is encouraging to see that the assumption we make about smart meters being capable of performing such aggregation is not unprecedented in the literature.

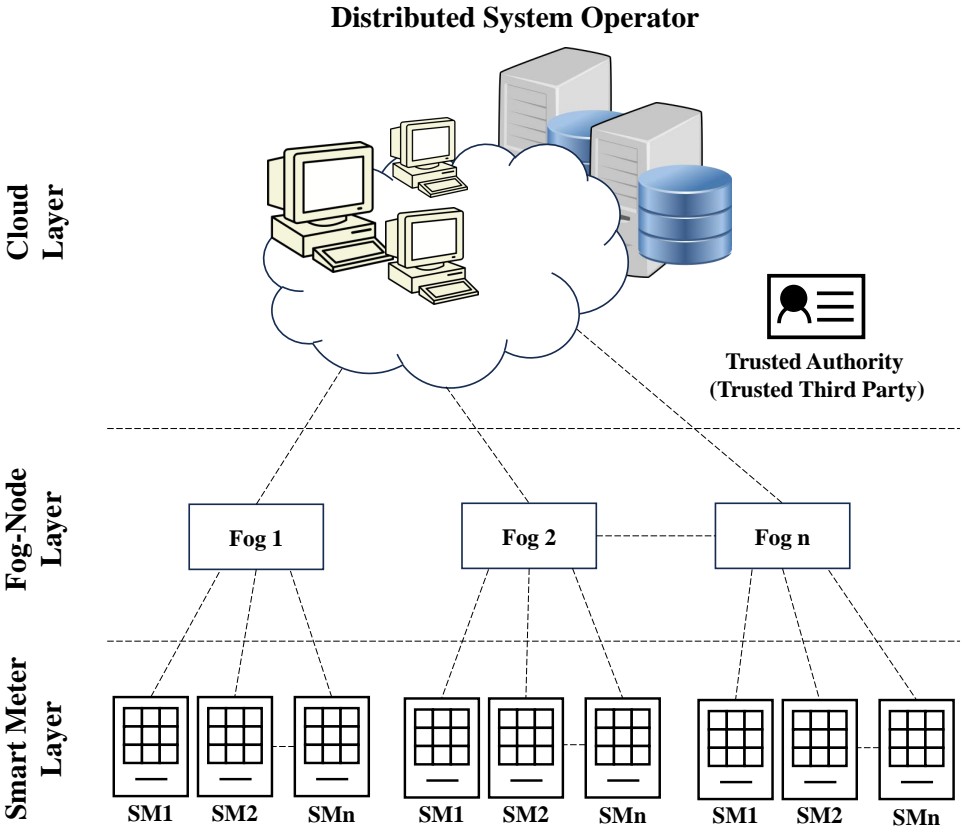

**Figure 1.** Typical architecture for an aggregation protocol employing fog nodes. (Figure adapted from Khan et al. [32]).

Table 2 presents the attributes of each of the relevant schemes.

**Table 2.** Comparison of related schemes (SA1: fog node independence; SA2: multidimensionality; SA3: collusion resistance; SA4: forward/backward secrecy; SA5: fault tolerance; SA6: dynamic membership; SA7: dynamic/variable group size; SA8: trusted third party/trusted authority independence).

| Scheme | SA1 | SA2 | SA3 | SA4 | SA5 | SA6 | SA7 | SA8 |
|--------|-----|-----|-----|-----|-----|-----|-----|-----|
| Liu et al. [37] | no | yes | yes | yes | yes | yes | no | no |
| Song et al. [39] | no | no | no | yes | no | yes | no | yes |
| Zuo et al. [40] | no | yes | yes | no | no | no | no | yes |
| Zhang and Liu [22] | no | yes | yes | yes | no | yes | yes | no |
| CHEA | yes | no | yes | yes | yes | yes | yes | yes |

Another example which shows conceptual similarities to our protocol is the solution proposed by Gomez Marmol et al. [41]. In their scheme, bilinear homomorphism is employed because it allows the aggregate of the data to have a separate decryption

key from the components that make up the aggregate. This way, the meters can send their encrypted data to the DSO directly (avoiding intermediate nodes), followed by an aggregate key, and the DSO can only decrypt the data once it has summed the ciphertexts (with some exceptions).

## 4. CHEA Scheme

CHEA is built around homomorphic encryption (HE), with algorithmic parameters described in Table 3. Although a specific HE implementation is not required to employ the protocol, for the purposes of implementation we have used the Paillier cryptosystem [27,28] due to its computational efficiency [31]. This last property is essential due to the relatively weak processing power of AMI equipment. In general, the only requirement is that the cryptosystem is additively homomorphic and supports unlimited additions (unless the group size parameter $\alpha$ is fixed at three, in which case it only needs to support two additions; in general, the HE must support $\alpha_{max} - 1$ additions, where $\alpha_{max}$ is some maximum acceptable value for $\alpha$. The parameters $\alpha$ and $\beta$ enable the DSO to dynamically adjust the level of aggregation as well as the precision of the aggregation regions. Zhang and Liu [22] note that variable subset sizes enable improved data analysis by the DSO. The capacity to target different regions and/or degrees of aggregation allows the scheme to support multiple privacy-preserving applications making use of aggregated data.

**Table 3.** Nomenclature.

| | |
|---|---|
| $N$ | number of aggregation groups |
| $G_n$ | $n^{th}$ aggregation group |
| $\alpha$ | group size parameter (must be three or above) |
| $\beta$ | locality parameter |
| $T$ | total $SM$ population |
| $SM_i$ | $i^{th}$ smart meter |
| $SM_j^{G_n}$ | smart meter at position $j$ in group $G_n$ |
| $SM_{leader}^{G_n}$ | leader of group $G_n$ (=$SM_0^{G_n}$) |
| $SM_x$ | currently selected smart meter |
| $P_i$ | $i^{th}$ selection pool |
| $L_i$ | leadership status of $i^{th}$ smart meter |
| $M_j$ | measurement of $SM_j^{G_n}$ |
| $A_j$ | aggregate at position $j$ in group (note: $A_0 = M_0$) |
| $Pk$ | public key of $SM_{leader}^{G_n}$ |
| $Vk$ | private key of $SM_{leader}^{G_n}$ |
| $E(X)Pk$ | cyphertext of $X$ encrypted using $Pk$ |
| $p, q$ | large prime numbers |
| $m, \gamma, \mu$ | cryptographic intermediaries |

### 4.1. Overview

CHEA works by dynamically splitting the region of SMs controlled by the DSO into distinct sets, referred to as groups. Each of these groups is a set of SMs that will have their data aggregated together. It is noted that different applications might require different densities of aggregation, or more or less tightly localized aggregation. In order to support these different applications, CHEA offers flexibility along these dimensions, expressed using the parameters $\alpha$ and $\beta$. $\alpha$ represents the number of SMs that will be in each group; a higher number equates to larger groups and less precise information, and vice versa. $\beta$ controls the localization of aggregation; a high $\beta$ value corresponds to a wide search region when generating groups, meaning that meters within groups may be physically distant. Conversely, a low $\beta$ value corresponds to a tight search region, meaning that meters within groups will be physically close (a necessity for some applications, such as demand response).

Once these parameters have been set, the DSO will generate a "plan" consisting of a set of groups and cycles within those groups. An example of this can be seen in Figure 2.

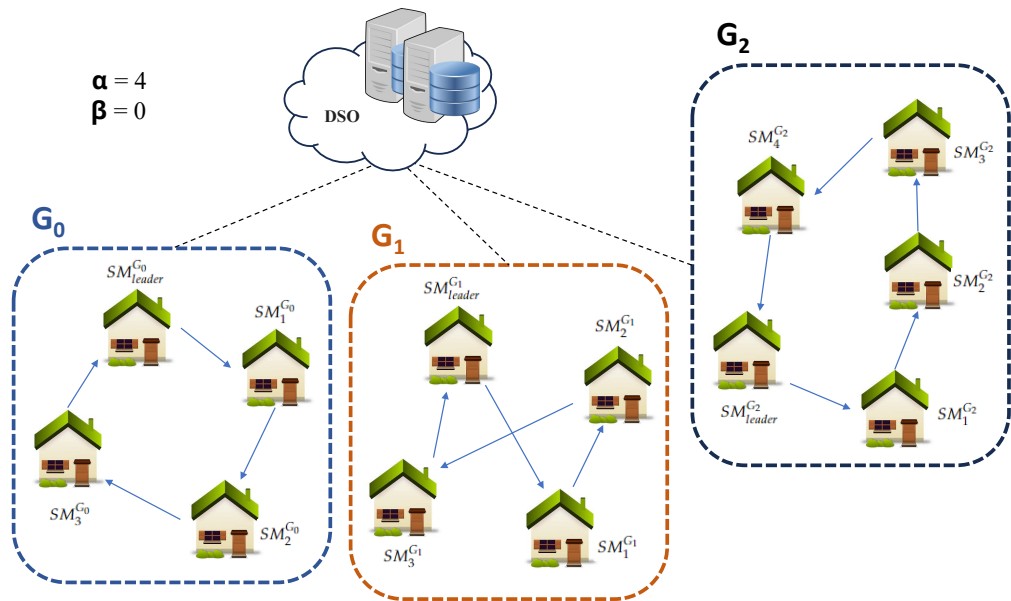

**Figure 2.** Sample CHEA plan construction at DSO is demonstrated with $T = 13, \alpha = 4, \beta = 0$.

Once the plan has been generated, the DSO sends a signal to each meter $SM_i$ to take a measurement along with a packet specifying their position in the group and the ID or address of the meter to which they will be sending data (Figure 3).

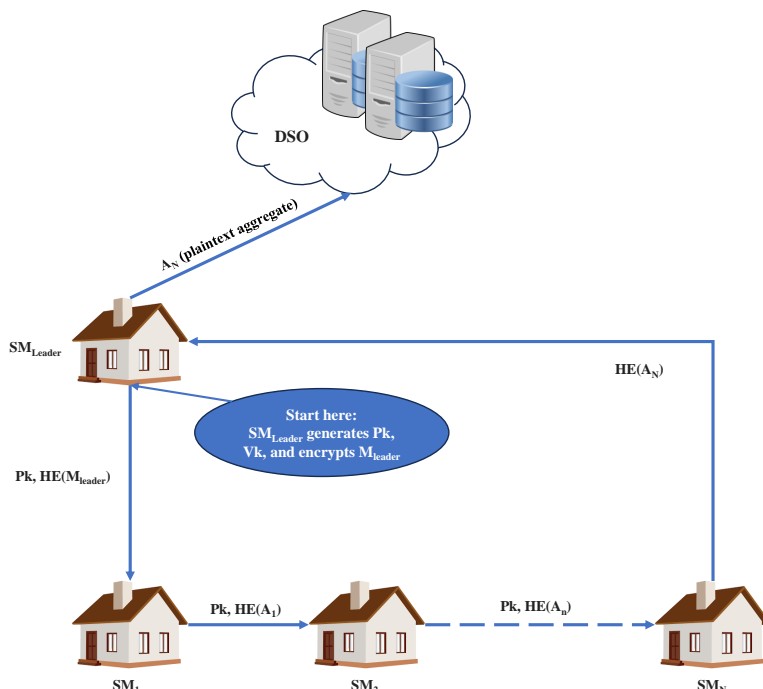

**Figure 3.** CHEA protocol aggregation phase is demonstrated; the dotted line represents an unknown number of intermediate nodes.

### 4.2. Formal Description

The aggregation process begins with the DSO initiating an aggregation request. The DSO will select values for the parameters $\alpha$ and $\beta$, which, respectively, determine the group

size (or data resolution) and locality. These parameters are set based on the application that the aggregated data will be used for.

At this stage, the DSO must generate aggregation groups of smart meters from within its population. It begins by sending a ping signal to all of the registered meters to ensure that only active meters are added to the plan. These active meters are split into regions of size $\beta^2$ using Algorithm 1. These pools are then used to generate groups of size $\alpha$ using Algorithm 2. While it cannot be guaranteed that a given pool will have a number of meters divisible by $\alpha$, the pools are constructed such that each subsequent pool will be geographically close to the previous pool. This means that, during Algorithm 2, the DSO can simply iterate over the pools to find the next closest meter if needed. This guarantees minimum deviation from the desired plan while maintaining computational efficiency.

---

**Algorithm 1** Beta Pool Creation

---

**Input:** $\beta$        ▷ size of aggregation regions
**Input:** $S = [SM_0, SM_1, ..., SM_n]$        ▷ list of smart meters
**Input:** $(Lat_{min}, Lon_{min}), (Lat_{max}, Lon_{max})$ ▷ DSO region boundaries (latitude, longitude)
**Output:** $[P_0, ..., P_{max}]$        ▷ list of pools
$i_{max} \leftarrow \lfloor Lat_{max}/\beta \rfloor$
$j_{max} \leftarrow \lfloor Lon_{max}/\beta \rfloor$
**for** $i = 0..i_{max}$ **do**
    **for** $j = 0..j_{max}$ **do**
        ▷ ensures that IDs are assigned in a snake-like pattern, i.e., subsequent pools will
        ▷ also be physically close
        **if** $i\%2 == 0$ **then**
            $pool.id \leftarrow i * j_{max} + j$
        **else**
            $pool.id \leftarrow i * j_{max} + j_{max} - j - 1$
        **end if**
        Initialize empty list $P_{pool.id}$
        Define $P_{pool.id}$ region as:
        $((Lat_{min} + \beta * i, Lon_{min} + \beta * j), (Lat_{min} + \beta * (i+1), Lon_{min} + \beta * (j+1))$
        Add all $SM_x$ to $P_{pool.id}$ where:
        $(Lat_{min} + \beta * i) \leq SM_x.Lat \leq (Lat_{min} + \beta * (i+1))$ **and**
        $(Lon_{min} + \beta * j) \leq SM_x.Lon \leq (Lon_{min} + \beta * (j+1))$
    **end for**
**end for**

---

Each group that is generated can be stored as a list of tuples, one for each smart meter, where the first half of the tuple is the SM identifier, and the second half of the tuple is the identifier points for the next SM in the group. For example, group $G_0$ would look like the following circular list: $[(SM_0^{G_0}, SM_1^{G_0}), (SM_1^{G_0}, SM_2^{G_0}), ..., (SM_{\alpha-1}^{G_0}, SM_0^{G_0})]$.

Additionally, a list containing the identifiers of the SM leader for each group is stored: $[(G_0, SM_{leader}^{G_0}), ..., (G_N, SM_{leader}^{G_N})]$

The DSO will now communicate this plan to the meters. Each meter receives a packet containing:

- The aggregation request.
- Its leadership status.
- The identifier/address of the next meter in the group (i.e., the one it will be sending data to).

Algorithm 3 is the algorithm used to generate public and private keys in the Paillier homomorphic cryptosystem. This algorithm will be run by the $SM_{leader}$ of each group in order to generate the public key, which will be sent to each meter in the group and used to encrypt each reading, as well as the private key, which will be used to decrypt the resulting aggregate before sending it to the DSO.

---

**Algorithm 2** Dynamic Group Generation

---

**Input:** $\alpha$      $\triangleright$ group size
**Input:** $T$      $\triangleright$ total population
**Input:** $P = [P_0, P_1, ..., P_{max}]$      $\triangleright$ list of pools
**Output:** $[G_0, ..., G_N]$      $\triangleright$ list of groups
$N \leftarrow \lfloor (T/\alpha) \rfloor$      $\triangleright$ number of groups
$j = 0$      $\triangleright$ pool index
**for** $i = 0..N - 1$ **do**
    Initialize empty list $G_i$
    **while** $P_j$ is empty **do**      $\triangleright$ find the earliest pool with ungrouped meters
        $j = j + 1$
    **end while**
    Select random $SM_x$ from $P_j$      $\triangleright$ select the group leader
    Add $SM_x$ to $G_i$ as $SM_{leader}$      $\triangleright$ $SM_0$ is the leader
    Remove $SM_x$ from $P_j$
    $Prev = SM_x$
    **if** $i < N - 1$ **then** $size \leftarrow \alpha - 1$      $\triangleright$ regular group size $= \alpha$
    **else** $size \leftarrow T - 1 - i * \alpha$      $\triangleright$ last group size $\geq \alpha$
    **end if**
    **for** $j = 1..size$ **do**      $\triangleright$ add the other group members
        **while** $P_j$ is empty **do**      $\triangleright$ find the earliest pool with ungrouped meters
            $j = j + 1$
        **end while**
        Select random $SM_x$ from $P_j$
        Add $SM_x$ to $G_i$ as $SM_j$
        Remove $SM_x$ from $P_j$
        $Prev.next = SM_x$
        $Prev = SM_x$
    **end for**
    $SM_x.next = SM_{leader}$
**end for**

---

**Algorithm 3** Paillier Key Generation (from Paillier [27])

---

**Output:** $Pk$      $\triangleright$ public key
**Output:** $Vk$      $\triangleright$ private key
Choose large primes $p, q \ni \gcd(p \times q, (p - 1) \times (q - 1)) = 1$ $\triangleright$ greatest common divisor
$m \leftarrow p \times q$
$\lambda \leftarrow \text{lcm}(p - 1, q - 1)$      $\triangleright$ least common multiple
Choose integer $g \in \mathbb{Z}_{m^2}^*$      $\triangleright$ such that g is relatively prime to $m^2$
$\mu = \left( \frac{m}{(g^\lambda mod(m^2)) - 1} \right) mod(m)$      $\triangleright$ modular multiplicative inverse
$Pk \leftarrow (m, g)$
$Vk \leftarrow (\lambda, \mu)$

---

Once the groups have been generated and the DSO has communicated these plans to the meters, aggregation can begin (see Algorithm 4). The full sequence of events, including the plan distribution, can be observed in terms of message exchanges in Figure 4.

### 4.3. Initialization

The DSO first selects values for $\alpha$ and $\beta$ based on the application that the data are being aggregated for (e.g., demand response, state estimation, etc.). The parameter $\alpha$ indicates the number of members in each aggregation group, and is thus inversely proportional to aggregation resolution. The parameter $\beta$, used in Algorithm 1, determines the maximum physical distance between smart meters in the same group.

**Algorithm 4** Aggregation

> **Input** Group $G_n[SM_0..SM_{z-1}]$       $\triangleright$ group of size $z \geq \alpha$
> **Output** $A_n$       $\triangleright$ aggregated value for $G_n$
> **for** $j = 0..z-1$ **do**       $\triangleright$ simultaneous
>      $M_j \leftarrow SM_j^{G_n}$ measurement       $\triangleright$ taking measurement
> **end for**
> $i \leftarrow 1$
> $SM_{leader} \leftarrow SM_0$
> $SM_{leader}$ generates public key $P_k$ and private key $V_k$ using Algorithm 3
> $SM_{leader}$ encrypt$(M_0)$ as $E(A_0)$       $\triangleright$ $A_0 = M_0$ in this case
> $SM_{leader} \rightarrow (SM_{leader}.next) : [E(A_0), Pk]$   $\triangleright$ send encrypted measure & key to next node
> $SM_i \leftarrow SM_{leader}.next$
> **while** $SM_i \neq SM_{leader}$ **do**
>      $SM_i$ encrypt$(M_i)$ as $E(M_i)$
>      $SM_i$ add$(E(M_i) + E(A_{i-1}))$ as $E(A_i)$
>      $SM_i \rightarrow (SM_i.next) : [E(A_i), P_k]$      $\triangleright$ send encrypted aggregated value & key to next
>      $SM_i \leftarrow SM_i.next$
>      $i \leftarrow i+1$
> **end while**
> $SM_{leader}$ decrypt$(E(A_{z-1}))$ as $A_n$
> $SM_{leader} \rightarrow DSO : A_n$

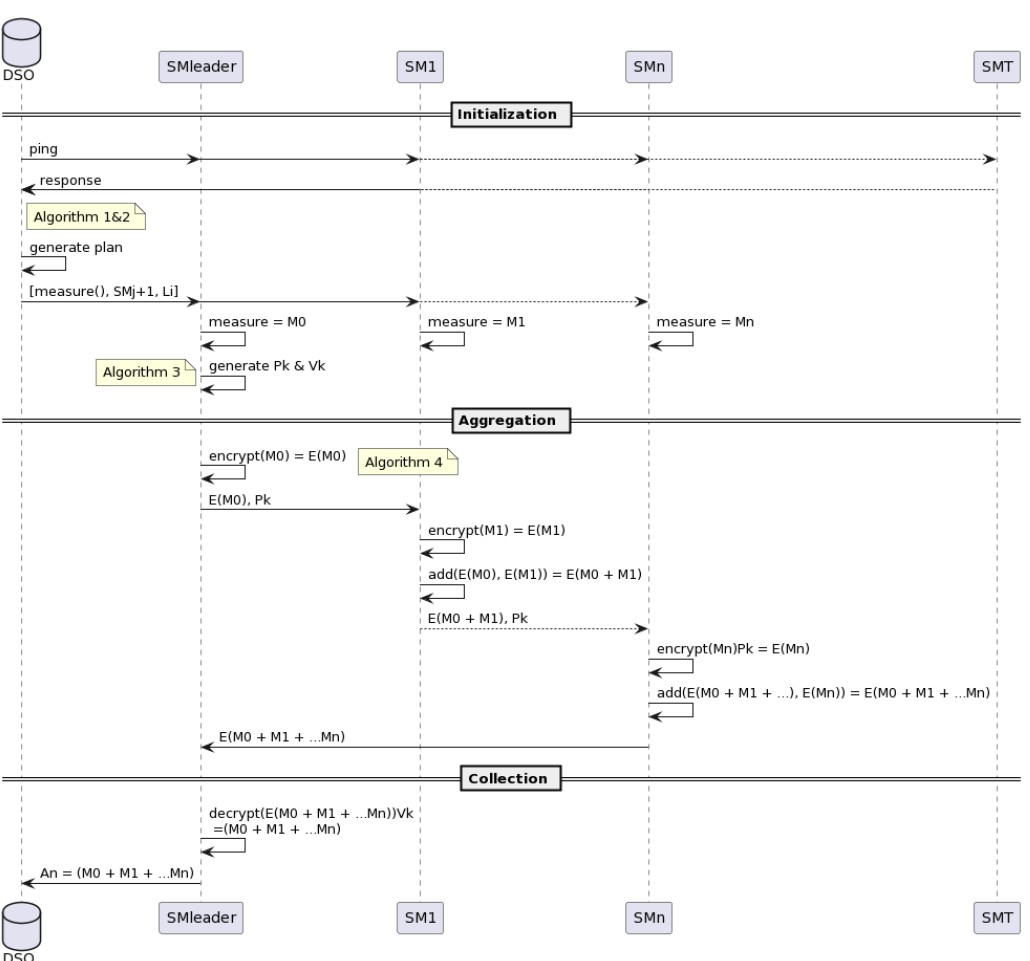

**Figure 4.** Execution cycle of the CHEA protocol with a group of n smart meters from a population of T smart meters illustrated as a sequence diagram. $SM_{leader}..SM_n$ are part of the group under focus but the others ($SM_T$) are part of other groups.

The DSO begins by generating $N = T/\alpha$ arbitrary groups of size $\alpha$ (except the last group, which can be larger), which is a tunable parameter based on the immediate needs of the DSO specified at the time of request. More specifically, Algorithm 2 generates a number of groups equal to the floor of the population divided by the group size. We ensure that groups contain at least three members each (as implied by the last comment in Algorithm 2). This is because we do not want to risk having a group of size two or one, which would remove the security benefit for those meters. In the case of one, it is clear that security cannot be attained via aggregation, and they would be relying purely on the security of the communication with the DSO, eliminating the benefit CHEA provides. In the case of a group size of two, the leader of the group would be able to determine the other meter's consumption data by subtracting their own data after the aggregation is received and decrypted.

Instead, we guarantee sufficient group size using the method described; as a result, all but one group are guaranteed to have size alpha, with the final group having $\alpha \leq size \leq 2 * \alpha - 1$.

In each of these groups, we identify one smart meter as the "leader", who begins the aggregation process, generates the required cryptographic keys, and reports the aggregated result to the DSO. Although all the meters will take their measurement simultaneously, the aggregation process necessarily takes place sequentially within each group.

### 4.4. Aggregation

The lead smart meter ($SM_{leader}^{G_n}$) will begin by generating a public and private key for the HE cryptosystem (Algorithm 3). It will then use this public key to encrypt its own measurement (Algorithm 4). The leader then sends its encrypted measurement along with the public key it generated to the next smart meter in the group (each meter will be sent the address of the next meter in the chain, the order of which is generated by the DSO at the same time the group is generated).

The next meter in the chain then encrypts its own measurement using the public key sent by the leader. By the property of additive homomorphism, it can add its encrypted measurement to the one sent by the leader, resulting in a cyphertext that contains the total usage by both the leader and the current meter.

As $\alpha$ is a minimum of three (to avoid privacy issues in small groups); the second meter sends the public key and current aggregate to a third meter, who then encrypts their own data using the private key and adds them to the current aggregate, creating a new encrypted aggregate that contains the sum of all three meters' usage. This goes on until the last meter of the group sends the cyphertext to the leader.

### 4.5. Decryption

The lead smart meter will have received the cyphertext containing the aggregate of the group's usage data from the last meter in the group. At this point, the leader will decrypt the usage data using the private key it generated at the beginning of the process. There is then no way for the leader to discover the usage data of any of the previous meters. This is because even if a leader maliciously decrypts the aggregate and subtracts its own usage data, the result will simply be the aggregate of the other $\alpha - 1$ meters' data, from which no precise information can be gained.

This aggregate data can now be sent to the DSO (end of Algorithm 4). Because the data are pre-aggregated, they can be sent in plaintext. However, the cost of encrypting with a public key from the DSO is insignificant and likely worth the small security improvement.

Note also that the groups are created dynamically and randomly at each request from the DSO to prevent statistical attacks that could be performed over multiple iterations.

*4.6. Faults*

Smart meters are sent a ping immediately prior to the generation of aggregation groups. This ensures that they are online and active before they are added to a group for a particular aggregation cycle (input *S* of Algorithm 2).

If groups are large enough ($\alpha$), this may not be a sufficient guarantee. In these cases, some communication efficiency can be sacrificed by including the full plan in each intermediate step, i.e., the full list of addresses in the group. Each meter in a given group would wait for a response from the subsequent meter before discarding the aggregate. If no response is received, they would defer transmission to the next address in the list.

## 5. Security Analysis

This section analyses how CHEA protects privacy under common attacks. This analysis goes beyond the points related to security, privacy, and fault tolerance made in the previous section.

*5.1. Man-in-the-Middle Attack*

A man-in-the-middle attack is a privacy attack where two parties are communicating and a third party, the attacker, intercepts this communication. The attacker may then use the intercepted information to facilitate burglary or blackmail, or may modify the information before it reaches its intended recipient.

A smart grid with TE running CHEA has several distinct interception points:

(a) Communication from the DSO to the *SM*s;
(b) Communication from the $SM_{leader}$ to the next *SM*;
(c) Communication from a non-leader *SM* to the next *SM*;
(d) Communication from the $SM_{leader}$ to the DSO.

In (a), the greatest risk would be altering the plan. The attacker may be able to change the address for the meter $SM_j^{G_n}$ to send information to. This would not enable false data injection (FDI), since they would need to intercept multiple transmissions in order to change the plan coherently; however, it could disrupt the cycle path and cause an aggregation failure for the group in question.

Communication from $SM_{leader}$ to the first intermediate meter in the group (b) poses a slight risk because it is the only transmission that contains non-aggregated data. However, it is encrypted using $SM_{leader}$'s public key, so they should be the only one capable of decrypting it. If they were to collude, they would only be putting their own data at risk.

Communication between two intermediate meters (c) poses little risk, since the data would be both encrypted and aggregated. Even in the unlikely scenario that the attacker can decrypt the message, it will not expose any private information.

Considering the final transmission (d), there is no privacy risk to the users in the aggregation region since the data have already been aggregated before this step occurs. The greatest risk would be FDI, which is the main reason the data should be protected using the public key of the DSO before sending.

*5.2. Collusion Resistance*

CHEA is extremely collusion resistant. Since group membership is allocated on the fly, it is impractical to form an adversarial coalition in advance. Additionally, an attacker would need the total aggregate that is sent *by* the victim (which may not be sent to them, depending on the plan), as well as the total aggregate that was sent *to* the victim. In the best case scenario, where the group size is three, this would require cooperation with one other user. Where $\alpha > 3$, it would require cooperation with two other users, and specifically the two users interacting with the victim (which is determined by the flow of the plan). Even if this contrivance was successfully achieved during one iteration, the plan would be extremely unlikely to succeed in the next iteration as group assignments and data flow would be reassigned.

Requirements ($SM_v$ is the victim):

- $SM_{leader}$ colludes;
- $SM_{v-1}$ colludes;
- $SM_{v+1}$ colludes;
- Attacker colludes or has (by chance) one of these three roles.

The probability of the DSO generating a plan that enables attackers to target a particular meter during a particular aggregation cycle is:

$$\frac{(\frac{\alpha}{N})^3 * (\prod_{i=1}^{3} \frac{i}{\alpha})}{(3!)^2}$$

where $N$ is the total smart meter population (or selection pool) and $\alpha$ is the group size during that cycle. The value three comes from the three parties required to collude with the attacker.

This expression can be simplified algebraically to:

$$\frac{1}{6N^3}$$

This function shrinks extremely quickly, demonstrating that the probability of successful collusion is negligible. With only 100 smart meters in a selection pool, the odds of success are already exceedingly low (1 in 6 million). In this case, even if aggregation was performed every second (an improbably high rate), a plan enabling a collusion attack would only be generated once every 70 days; this frequency is not useful for any class of cyberattack.

### 5.3. Quantum Attacks

Although our implementation and simulation employed the Paillier encryption scheme, the core CHEA protocol is cryptosystem agnostic. Therefore, it would be trivial to implement the protocol with a quantum-resistant cryptosystem, such as lattice-based cryptography [42,43].

## 6. Performance Analysis

### 6.1. Overview

We simulated the operations carried out by the smart meters and the DSO, respectively, using two custom C programs that communicate using sockets, as they would in a real networking application. Simulations were performed on a 2017 MacBook Pro with a Intel Core i5 CPU at 3.1 GHz and 8 GB RAM. The code is available online in a replication package [44].

Each smart meter process simulates the operations carried out by a smart meter, and contains a unique identity that comprises physical variables associated with the smart meter being simulated to improve the accuracy of the simulation. The DSO application acts as a server for the smart meters and generates and distributes the plan as described in Section 4.

In addition to employing a realistic network implementation, we further enhanced the simulation's accuracy by employing current IEEE smart meter communication protocols as outlined by Zaraket et al. [45] and Shanmukesh et al. [46].

### 6.2. Results

We compared the performance of our scheme against schemes by Liu et al. [37], Song et al. [39], and Zuo et al. [40]. Table 4 summarizes the results in terms of net communication overhead for each scheme. Table 5 summarizes the results of processing time impacts for each phase of aggregation for each scheme.

Processing time for competing schemes was produced by simulating several tests of the most impactful operations on our hardware and extrapolating the assessment method employed by Liu et al. [37]. The Stanford Pairing-Based Cryptography library was em-

ployed to perform the tests, which were run on the aforementioned 2017 MacBook Pro. The testing parameters were *population* = 30 and *dimensionality* = 10. CHEA was extended theoretically to support multidimensionality to ensure a fair comparison with the other schemes.

**Table 4.** Comparison of communication overhead.

| Scheme | Initialization Phase | Aggregation Phase | To DSO |
| --- | --- | --- | --- |
| CHEA | 1024 bits | 2072 bits | 1024 bits |
| Liu et al. [37] | 3456 bits | 2368 bits | 2368 bits |
| Song et al. [39] | 4192 bits | 1056 bits | 1024 bits |
| Zuo et al. [40] | 1088 bits | 1600 bits | 1600 bits |

**Table 5.** Comparison of performance (times averaged over 10 runs)

| Scheme | Initialization Phase | Aggregation Phase | Decryption Phase |
| --- | --- | --- | --- |
| CHEA | 125 ms | 497 ms | 29 ms |
| Liu et al. [37] | 79 ms | 340 ms | 27 ms |
| Song et al. [39] | 122 ms | 1036 ms | 34 ms |
| Zuo et al. [40] | 119 ms | 731 ms | 1337 ms |

We found that our scheme has communication overheads comparable to the other schemes. Specifically, our initialization phase and end phase are as good as or better than the schemes compared. In the aggregation phase, our scheme has the second highest communication overhead. This result was anticipated, as sending the public key ($P_k$) along with the intermediate aggregate ($E(A_i)$) necessarily incurs additional communication cost.

With regards to processing time, we found that our scheme is comparable to the others in the decryption phase, except for the much slower approach from Zuo et al. [40]. This result is not surprising since there is no additional processing required. CHEA's initialization phase is the slowest of the group (but not by much); this can be attributed to the extra processing required to generate the plan each round. In the aggregation phase, CHEA is the second fastest. It is slower than Liu et al. [37]'s scheme because encryptions must be performed sequentially (other schemes have a parallel reporting phase). However, the structure of the scheme requires fewer cryptographic operations overall, resulting in performance gains compared to the rest of the schemes. Additionally, scaling analyses indicate that our scheme could see a comparably superior performance as neighbourhood sizes increase, as we note and explain in the next section.

Overall, our scheme incurs a slight communication disadvantage compared to *some* similar schemes, as well as a slower initialization time. However, the aggregation time is competitive and scaling (discussed in Section 6.3) is improved. Additionally, CHEA improves flexibility, fault tolerance, and security. For example, although Liu et al. [37] remains the fastest scheme, it lacks the flexibility of CHEA's dynamic group membership, which offers versatile locality and aggregation resolution.

### 6.3. Scaling

**Registration.** Since there is no centralized key distribution, the registration/initialization phase is essentially of complexity $\mathcal{O}(1)$, incurring no extra time cost as the population of smart meters grows. As we discuss in Section 1, local in-group key generation, the distributed nature of the protocol, and the avoidance of complex key distribution make our scheme significantly more scalable than comparable protocols [39,47].

**Aggregation.** As is the case with registration (Section 6.2), the aggregation happens locally and concurrently within each group, thus there is no time scaling associated with having larger neighbourhoods (i.e., more groups). Larger groups (corresponding to a larger $\alpha$ value), on the other hand, *will* incur an increased time penalty during this phase, since the aggregation is generated sequentially within a group.

Scaling tests were performed using the custom simulation, the results of which are presented in Figure 5. The simulation used libhcs instead of PBC for HE, hence the faster times relative to the earlier test. For these tests, $\beta$ was set to zero (which is interpreted as nearest neighbour instead of pooling) and the neighbourhood size was 10 km$^2$. Population and $\alpha$ were varied as seen in the graph.

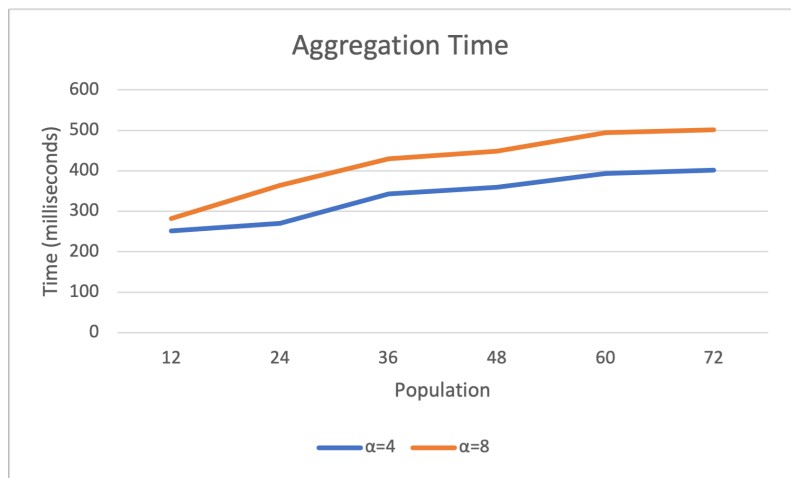

**Figure 5.** CHEA aggregation times for different neighbourhood populations. The sublinear increase can largely be explained by the computer being under greater load as it simulates more smart meters.

It can be seen that the results support the theoretical scaling properties; that is, increasing the population has a limited effect on performance, while increasing $\alpha$ has a small but noticeable effect. Unfortunately, simulating all devices (and, in particular, DSO operations) on the same computer somewhat dampens the visibility of these effects, and future tests should include a distributed hardware simulation to better isolate them.

## 7. Limitations

Due to the lack of available hardware for testing, the protocol was tested using a software simulation of a smart grid environment. Ideally this would have been tested in a more accurate setup using real, physical smart meters to demonstrate feasibility and gather data for prospective users. However, given that most current-generation AMIs are not capable of performing the necessary operations, it is difficult to test HE-based aggregation schemes in practice at this time. As a result, related literature largely relies on software simulations, as we did.

Another limitation lies in the assumptions made about the capabilities of smart meters. Although we did not make any unprecedented assumptions—that is, assumptions that have not already been made in similar literature—we do presume smart meters to be capable of certain tasks that are not currently among their commonly-found functionalities. This may call into question the feasibility of this architecture as a real-world solution. In discussions with industry experts, it was noted that manufacturers prefer to avoid unnecessary upgrades to grid hardware, which may limit short-term adoption. However, in spite of the current state of smart meter hardware, it seems reasonable to assume that they will follow the trend of computational ability in virtually every other domain, and that they will receive increased computing power at decreased costs as time goes on. As this happens, it will become cost effective for utility companies to provide increased processing capabilities to metering equipment in order to support more comprehensive security measures, as this will increase consumer trust, provide marketing opportunities, and reduce expensive breaches, all at limited cost. For these reasons, we believe security research regarding the smart grid should not be limited to current hardware capabilities, and continue the example set by other researchers with CHEA.

## 8. Conclusions and Future Work

This paper introduced Cyclic Homomorphic Encryption Aggregation (CHEA) as a new scheme for protecting privacy. We found CHEA to be effective in providing privacy-protecting aggregation of energy usage data for distributed smart grid energy trading settings, including TEMs. Formal analysis and software simulation confirm that the protocol provides significant security benefits without sacrificing performance.

Although the scheme performs similarly to other smart grid aggregation schemes, current smart meters are likely incapable of supporting the requisite cryptographic operations, so real-world deployment will depend on hardware improvements in AMI.

The CHEA protocol presents a novel, distributed, HE-based aggregation solution for TE that could potentially be generalized to other environments with similar infrastructures, e.g., environments consisting of networked devices that generate data to be consolidated, and operate on distributed applications. One potential candidate may be smart electric vehicles, which generate driving quality and accident data to inform insurance providers for different demographics. Other metered utilities, such as water and gas, may also potentially benefit from our solution, although extending the scheme to these areas may present novel domain-specific challenges.

Some other potential areas for future research include:

- Improving upon CHEA by making it even more robust against communication or meter failures during the aggregation phase. Future iterations could include plans that are dynamically adjusted based on where communication drops off, but this will require making the meters even more autonomous (thus increasing their computational load).
- Investigating different methods of group generation. It may be possible to forgo the requirement of the DSO creating a plan centrally if smart meters create the plan dynamically in a more procedural manner, perhaps using cellular automata, for example. While there is no guarantee that this would be more efficient, it could be an interesting research topic and would at least confer the benefit of increasing distribution, reducing reliance on centralized computing even further.
- Adding finer control to the locality parameter to enable the DSO to be more specific about how regions are divided. For example, it may want to consider network topology, neighbourhoods, or other currently unsupported factors when requesting an aggregate reading.
- Looking into other applications of HE to grid management operations. This technology may enable distributed applications for functionality such as state estimation, transaction verification, or power flow optimization. Distributing these tasks could present further privacy, security, and reliability benefits to TEM users and operators, similar to those seen with CHEA.

**Author Contributions:** Conceptualization, A.R.-K., D.A., D.S.-D. and J.M.; methodology, D.S.-D. and D.A.; software, D.S.-D.; validation, D.S.-D. and M.B.; formal analysis, D.S.-D.; investigation, D.S.-D.; data curation, D.S.-D.; writing—original draft preparation, D.S.-D. and D.A.; writing—review and editing, A.R.-K., J.M., D.S.-D., M.B. and D.A.; supervision, D.A., J.M. and A.R.-K.; project administration, D.A., J.M. and A.R.-K.; funding acquisition, D.A., J.M., M.B. and A.R.-K. All authors have read and agreed to the published version of the manuscript.

**Funding:** This research was funded by the ORF-RE project *CyPreSS: Software Techniques for the Engineering of Cyber-Physical Systems* as well as an NSERC Discovery Grant titled *Engineering Requirements for Socio-Technical Systems*.

**Institutional Review Board Statement:** Not applicable.

**Informed Consent Statement:** Not applicable.

**Data Availability Statement:** The simulation code is freely available online [44].

**Acknowledgments:** The authors thank Luigi Logrippo for feedback on this work and for providing pointers to useful references. We also thank Javad Fattahi for taking the time to examine this project and sharing his expertise with us, as well as the anonymous reviewers for their constructive feedback.

**Conflicts of Interest:** Authors Ashkan Rahimi-Kian and Masoud Bashari were employed by the company IEMS Solution Ltd. The remaining authors declare that the research was conducted in the absence of any commercial or financial relationships that could be construed as a potential conflict of interest.

## Abbreviations

The following abbreviations are used in this manuscript:

| | |
|---|---|
| AMI | Advanced Metering Infrastructure |
| BESS | Battery Energy Storage System |
| CHEA | Cyclic Homomorphic Encryption Aggregation |
| DER | Distributed Energy Resource |
| DSO | Distribution System Operator |
| EV | Electric Vehicle |
| FDI | False Data Injection |
| FN | Fog Node |
| HE | Homomorphic Encryption |
| ICT | Information and Communication Technology |
| P2P | Peer-to-Peer |
| RES | Renewable Energy Sources |
| SG | Smart Grid |
| SM | Smart Meter |
| SPOF | Single Point of Failure |
| TA | Trusted Authority |
| TE | Transactive Energy |
| TEM | Transactive Energy Market |
| TTP | Trusted Third Party |

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
