# Peer review of "Cyclic Homomorphic Encryption Aggregation (CHEA)—A Novel Approach to Data Aggregation in the Smart Grid"

_energies, doi:10.3390/en17040878_

Round 1

Reviewer 1 Report

Comments and Suggestions for Authors

Comments and Suggestions for Authors

This paper introduces a secure aggregation protocol Cyclic Homomorphic Encryption Aggregation (CHEA), that eliminates the need for additional network components or complicated key distribution schemes, while providing additional capabilities compared to similar protocols. Verification results indicate that CHEA has many desirable security properties and performs well in comparison to similar works, with minimal communication overhead, and is a strong candidate for aggregating transactive energy data.

I would like to suggest revisions that I believe would further improve the quality and clarity of the manuscript:

1. In the abstract, a comparison is made between CHEA and existing methods, highlighting the advantages of CHEA. These results are meaningful; however, specific numerical values are lacking. It is recommended to briefly list these performance metrics in the abstract to provide readers with a more intuitive understanding of the improvements.

2. The paper mentions the existence of several solutions, but it does not provide a detailed explanation of the working principles and pros and cons of these solutions. It is advisable to conduct a more in-depth analysis and comparison of these solutions in the paper to help readers better understand the research background and contributions.

3. Various methods for processing energy usage data are mentioned in the paper, such as algebraic transformations, data aggregation, and random noise. It is recommended to provide more detailed explanations of the working principles and advantages and disadvantages of each method in the paper to help readers better understand the application scenarios of these methods.

4. The paper mentions the use of the Paillier homomorphic encryption system but does not provide information about why this system was chosen and comparisons with other homomorphic encryption systems. It is suggested to provide more detailed information about the implementation of homomorphic encryption and the rationale behind the selection in the paper.

5. The paper mentions the adjustment of parameters α and β but requires more information to explain how to determine the optimal parameter values for different application scenarios. It is recommended to provide guidance on parameter selection and examples of practical applications.

6. The paper mentions the use of two custom C programs for simulation experiments but needs to provide more detailed descriptions of the experiments so that readers can understand the specific process and settings. For example, more information can be provided about simulation parameters, input data, and experimental procedures.

7. The paper mentions performance comparisons with other solutions but needs to provide a more detailed description of the performance comparison metrics and methods. Readers need to understand how performance was measured and compared to evaluate the results.

8. In the performance results section, more explanations are needed to help readers understand why certain aspects of performance are poorer and how these differences affect the practical application of the protocol.

9.The authors think that the research of this paper cannot be considered strictly a performance improvement, but it improved flexibility, fault tolerance, and security.Please provide a detailed description of where improvements in flexibility, fault tolerance, and security are reflected.

10.In Section 6, the simulation results were compared with the results of the references in Tables 2 and 3. Please supplement whether the testing environment tested in this article is consistent with the testing environment in the references.

11.In Section 6, the DSO application acts as a server for the smart meters and generates and distributes the plan. Please provide information on how the parameters in the relevant algorithms are determined.

12. The paper mentions the scalability of the protocol but needs to discuss the practical significance and impact of this feature in more detail. Readers need to understand how the protocol adapts to different scales of smart meter networks.

13. The paper mentions that due to the lack of available hardware for testing, software simulation was used to conduct tests. However, the best approach is to test with real physical smart meters in a real environment. It is advisable to discuss more explicitly in the paper the impact of the lack of hardware testing and possible solutions.

In short, I believe your research is valuable, but it requires significant revision and improvement.

Author Response

Thank you for your comments. Please see attached PDF file for answers.

Reviewer 2 Report

Comments and Suggestions for Authors

This paper is very well structrured and presented very clearly. The results are given and clarified. The only question is if the results presented in Table 3 can be improved, but considering the conclusions part, there is a lot of that is planned to be improved. 

Author Response

(The authors gave the same response as above.)

Reviewer 3 Report

Comments and Suggestions for Authors

The CHEA proposition is worth to be further investigated, publication of the current paper is therefore recommended.

In further work, the following areas are suggested:

  • A larger scale simulation

  • A simulation comparison with a blockchain equivalent

  • A trial

And 

  • The possibility to extend to other than electricity grid infrastructure, e.g. for water distribution, using the electricity grid AMI as a hub

It is noted that the implementation of CHEA requires more powerful AMIs

The main issues with AMIs are twofold:

  • Power consumption, in case of battery operation

  • Computing power

For electricity grid applications, which may be or become the most important for modern society, the power issue is minimal, as no permanent battery operation has to be taken into account. Leaving the increased computing power as a potential issue (however, it may be assumed that AMIs designed for the electricity grid may have a certain computing power reserve, as power consumption critical design tends to be more costly.

The possible extension to other distribution infrastructure, in particular water, but possibly also gas, is a challenging prospect (whereas gas may be on tis way out, the measurement of consumption may therefore be even more critical)

Author Response

(The authors gave the same response as above.)

Reviewer 4 Report

Comments and Suggestions for Authors

This paper introduced Cyclic Homomorphic Encryption Aggregation (CHEA), a secure aggregation protocol that eliminates the need for additional network components or complicated key distribution schemes, while providing additional capabilities compared to similar protocols. But the following problems still exist, and the author is suggested to revise them carefully.

1. In the instruction, please summarize the main innovative points of this paper again, for example: The detailed contributions of this study are summarized as follows:1.Presents a xxx method, which can xxxxxxxxxxx. 2.xxxxxxxxxx. 3.xxxxxxxxxxxxxx.

2. Suggest integrating the content of 2. Background and Motivation and 3. Related Work into one chapter, both of which are discussing the background.

3. Are Tables 2 and 3 three line tables? It is suggested to revise them carefully.

4. In this paper, the protocol was tested using a software simulation of a smart grid environment, What software is being used? How are the parameters set? Can you explain the effectiveness of the algorithm in this article?

5. The transactive energy market is an emerging development in energy economics built on advanced metering infrastructure. Data generated in this context is often required for market operations, while also being privacy sensitive. Therefore, for this topic, data encryption algorithms are very important. Suggest the author to cite some of the latest relevant research literature. Such as:

[1] Kou, L.; Wu, J.; Zhang, F.; Ji, P.; Ke, W.; Wan, J.; Liu, H.; Li, Y.; Yuan, Q. Image encryption for Offshore wind power based on 2D-LCLM and Zhou Yi Eight Trigrams. International Journal of Bio-Inspired Computation. 2023, 22(1): 53-64. https://doi.org/10.1504/IJBIC.2023.133505

[2] O. M. Al-Hazaimeh, M. F. Al-Jamal, A. K. Alomari, M. J. Bawaneh, and N. Tahat, “Image encryption using anti-synchronisation and Bogdanov transformation map,” International Journal of Computing Science and Mathematics, 2022, vol. 15, no. 1. pp. 43–59. https://doi.org/10.1504/IJCSM.2022.122144

[3] Cheng, Y.; Liu, Y.; Zhang, Z.; Li, Y. An Asymmetric Encryption-Based Key Distribution Method for Wireless Sensor Networks. Sensors 2023, 23, 6460. https://doi.org/10.3390/s23146460

Author Response

(The authors gave the same response as above.)

Round 2

Reviewer 4 Report

Comments and Suggestions for Authors

none